# Fifty Shades of Scandium: Comparative Study of PET Capabilities Using Sc-43 and Sc-44 with Respect to Conventional Clinical Radionuclides

**DOI:** 10.3390/diagnostics11101826

**Published:** 2021-10-03

**Authors:** Thiago V. M. Lima, Silvano Gnesin, Klaus Strobel, Maria del Sol Pérez, Justus E. Roos, Cristina Müller, Nicholas P. van der Meulen

**Affiliations:** 1Department of Radiology and Nuclear Medicine, Luzerner Kantonsspital, 6004 Luzern, Switzerland; klaus.strobel@luks.ch (K.S.); marisol.perez@luks.ch (M.d.S.P.); justus.roos@luks.ch (J.E.R.); 2Institute of Radiation Physics, Lausanne University Hospital, University of Lausanne, 1011 Lausanne, Switzerland; silvano.gnesin@chuv.ch; 3Center for Radiopharmaceutical Sciences ETH-PSI-USZ, Paul Scherrer Institute, 5232 Villigen, Switzerland; cristina.mueller@psi.ch (C.M.); nick.vandermeulen@psi.ch (N.P.v.d.M.); 4Laboratory of Radiochemistry, Paul Scherrer Institute, 5232 Villigen, Switzerland

**Keywords:** PET, quantification, novel radionuclides

## Abstract

Scandium-44 has been proposed as a valuable radionuclide for Positron Emission Tomography (PET). Recently, scandium-43 was introduced as a more favorable option, as it does not emit high-energy γ-radiation; however, its currently employed production method results in a mixture of scandium-43 and scandium-44. The interest in new radionuclides for diagnostic nuclear medicine critically depends on the option for image-based quantification. We aimed to evaluate and compare the quantitative capabilities of scandium-43/scandium-44 in a commercial PET/CT device with respect to more conventional clinical radionuclides (fluorine-18 and gallium-68). With this purpose, we characterized and compared quantitative PET data from a mixture of scandium-43/scandium-44 (~68% scandium-43), scandium-44, fluorine-18 and gallium-68, respectively. A NEMA image-quality phantom was filled with the different radionuclides using clinical-relevant lesion-to-background activity concentration ratios; images were acquired in a Siemens Biograph Vision PET/CT. Quantitative accuracy with scandium-43/scandium-44 in the phantom’s background was within 9%, which is in agreement with fluorine-18-based PET standards. Coefficient of variance (COV) was 6.32% and signal recovery in the lesions provided RC_max_ (recovery coefficient) values of 0.66, 0.90, 1.03, 1.04, 1.12 and 1.11 for lesions of 10-, 13-, 17-, 22-, 28- and 37-mm diameter, respectively. These results are in agreement with EARL reference values for fluorine-18 PET. The results in this work showed that accurate quantitative scandium-43/44 PET/CT is achievable in commercial devices. This may promote the future introduction of scandium-43/44-labelled radiopharmaceuticals into clinical use.

## 1. Introduction

[^18^F]fluoro-deoxy-glucose ([^18^F]FDG) is still predominant in PET imaging, but there is large interest in the development of further radionuclides to benefit from more favorable decay characteristics to match specific applications. In this regard, the availability, transport, dosimetry, and safety are critical aspects to be addressed [1,2,3,4,5].

Previous work with scandium-44 showed the capability of clinical systems to deal with non-pure emission radioisotopes provided the appropriate corrections were applied [6,7]. Despite the promising results obtained with scandium-44 [6,7], the emission of high-energy γ-radiation (Eγ = 1157 keV) may be considered a drawback. This opens the discussion of whether another radioisotope from the Sc family, namely scandium-43, with similar physical characteristics but without the emission of high-energy γ-rays, would be more suitable for clinical applications.

There are different routes of production of scandium-43 in a cyclotron [8]. The production of scandium-43 is challenging due to the cost of target material and generated yield [8]. A new approach, simplifying the production based on scandium-44 production, was proposed, but although the yield was sufficient the level of purity was linked to the cross-section of proton beam on target. As a result, the irradiation of enriched calcium-44 targets using the (p,2n) nuclear reaction would result in a mixture of scandium-43 and scandium-44 [9].

A mixture of radioisotopes would not affect the radio labelling capability, and as the pharmacokinetic properties of the radiopharmaceutical are dependent on the chemical characteristics, ^44^Sc- and ^43^Sc-labeled biomolecules would result in exactly the same tissue distribution profile. The use of a mixture of scandium-43 and scandium-44 could, thus, be a valid option for future clinical application, particularly as both radioisotopes have similar half-lives.

Newer-generation SiPM PET/CT has shown improved sensitivity, count rate and time-of-flight performance with respect to conventional PMT PET/CT, with a potentially favorable impact in terms of quantification of non-pure gamma emitters such as scandium-44.

The aim of this work was, therefore, to compare and characterize the quantitative capabilities of a latest-generation clinical SiPM PET/CT for a mixed application of scandium-43/scandium-44 versus radioisotopically pure scandium-44, as well as the clinically used fluorine-18 and gallium-68.

## 2. Materials and Methods

### 2.1. Phantoms and Devices Used for the Measurements

In order to evaluate quantification capabilities and image quality of several radionuclides, a NEMA IQ phantom (Pro-Project) filled with the radionuclide in question was acquired on our recently installed Biograph Vision (Siemens Healthineers, Erlangen, Germany), with a step-and-shot acquisition covering the whole extend of the phantom in one bed position. The phantom photo and characteristics are given in Figure 1 and Table 1, respectively. Acquisitions were made in list mode and for 300 s duration.

PET/CT data were reconstructed using the clinical reconstruction algorithm (TrueX.HD + TOF), including point spread function and time-of-flight using four iterations and five subsets, all pass filter and 440 matrix size. For comparison with the literature [7], both absolute (aSC) and relative (rSC) scatter corrections were used for both the application of radioisotopically pure scandium-44, as well as the radioisotope mixture of scandium-43 and scandium-44.

### 2.2. Radionuclides and Phantom Activity Concentrations

In this study, the scandium-43 and scandium-44 (68% scandium-43 and 32% scandium-44) mixture was compared to radioisotopically pure scandium-44, fluorine-18 and gallium-68, respectively. Physical properties for the different radionuclides are described in Table 2 below. The ratio between scandium-43 and scandium-44 was the best result achievable under the current production conditions using enriched calcium-44 target material. For each isotope or radioisotope mixture, NEMA phantom PET acquisitions were performed on different days to avoid radioactive contamination between experiments.

In these experiments, the aim was to obtain a background activity concentration of ~5 kBq/mL and a lesion-to-background ratio of 5:1. Table 3 describes the radionuclide concentrations used in the sphere and in the main background (bkg) volume of the NEMA phantom.

### 2.3. Local Calibration of Dose Calibrator

For both scandium-44 and the scandium-43/44 mixture, two different approaches were used to obtain a local calibration factor for the dose calibrator (Veenstra VIK 202) at Luzerner Kantonsspital, Luzern, Switzerland. The first approach was based on a linearity method, where several measurements were performed by varying the calibration factor. The second approach employed, based on the vendor recommendations, calculated calibration factors based on the known reference activity and the activity measured with the technetium-99m factor using the following equation:Calibration factor_radioisotope_ = 1009 − (V_radioisotope_/V_Tc-99m_) × 773,(1)
where V_radioisotope_ is the reference activity for scandium (either scandium-44 or scandium-43/44 mixture) and V_Tc-99m_ is the measured vial using the factor for technetium-99m.

The linearity method was applied to obtain the calibration factor, while the vendor approach was used for validation of the calibration factor determined with the linearity method. Reference activities were 1406 MBq and 870.7 MBq for scandium-43/scandium-44, respectively. Calculations with both methods considered decayed corrected activities to their respective calibration times. The factors obtained were applied to all activity measurements used to generate the activity concentrations present in the phantoms.

For both fluorine-18 and gallium-68, calibration factors are restricted by the federal institute for metrology (METAS), which defines that calibration factors for clinically used radionuclides may only be modified under the responsibility of the manufacturer. After a change in the calibration factors, the dose calibrator must be recalibrated before the next use and METAS must be notified of the changed calibration factors [10].

### 2.4. Phantom Analysis

#### 2.4.1. PET vs. Dose Calibrator Activity Cross-Calibration (*BGcal*)

The PET-to-local-dose-calibrator activity cross calibration (*BG_cal_*) was tested by calculating the ratio between the measured PET activity concentration in the NEMA phantom background (${\bar{A}}_{c,bg}$) and the expected average activity concentration (*A_c_,_bg_*) in the same volume, known by the experiment construction:(2)$$BG_{cal}=\frac{{\bar{A}}_{c,bg}}{A_{c,bg}},$$

${\bar{A}}_{c,bg}$ was measured in three spherical regions of interest (50 mm in diameter) placed in the homogeneous background region of the phantom that surrounded the spheres. When using [^18^F]FDG, a deviation <0.1 from the ideal *BG_cal_* = 1 is regarded as acceptable.

#### 2.4.2. Image Noise

The image noise was evaluated by the coefficient of variation (*COV*), which is the ratio between the standard deviation and the average activity concentration measured in the same volumes of interest (VOIs) of the phantom background described above for the cross-calibration assessment:(3)$$COV\;\left(\%\right)=\frac{SD_{bg}}{{\bar{A}}_{c,bg}}\;\times \;100,$$

The background signal-to-noise ratio (SNR) is the reciprocal of the *COV*. A *COV* ≤15% (background SNR ≥6.7) was considered as an acceptable noise level for clinical image interpretation, as suggested in the EARL procedure [11]. Although this value is arbitrary, it has already been used as a reference value in previously published work [12,13,14], which enables a term of comparison for image-quality assessments.

#### 2.4.3. Average Residual Lung Error

A cylindrical VOI, 35 mm in diameter and 120 mm in length, was included in the lung insert. The average residual lung error (in %) was obtained by:(4)$$\bar{LE}\;\left(\%\right)=\frac{{\bar{L}}_{c}}{{\bar{A}}_{c,bg}},$$
where ${\bar{L}}_{c}$ is mean counts in the lung region and ${\bar{A}}_{c,bg}$ the average activity concentration in the background region.

#### 2.4.4. Recovery Coefficients

VOIs were manually segmented on the CT component of the acquisition, matching the sphere volumes included in the NEMA phantom. For each spherical insert (*j* = 1,…6) the maximum and the mean recovery coefficients (*RC*) were obtained as follows:(5)$$RC_{j,X},\;(\%)=\frac{a_{c,j,X}}{A_{c,sph}}\;\times \;100$$
where A*_c_,_sph_* is the expected activity concentration in the spheres, $a_{c,j,X}$ is the measured activity concentration for a given spherical insert (*j*) for either the mean and the maximum evaluations (*X*). Calculated *RC_max_* and *RC_mean_* values were compared with reference values provided by the EANM/EARL accreditation protocol for F-18 PET [11,15].

### 2.5. Software

All segmentation and image assessments were done using clinical MIM workstation Version 7.0.5. Analyses were performed with Microsoft Excel.

## 3. Results

### 3.1. Local Calibration of Radionuclides

#### 3.1.1. Linearity Method

Figure 2 shows the measured activity obtained for different calibration factors for the scandium-43/44 mixture and pure scandium-44, respectively. Measurements were decay-corrected to the reference time.

Calculated calibration factors based on the linearity measurements were determined as 836 scale 1 and 867 scale 1, respectively, for the radioisotopic mixture of scandium-43/44 and the radioisotopically pure scandium-44. With the factor obtained for scandium-43/44, an activity of 1419 MBq was measured, which corresponded to a 0.89% deviation from the reference activity. With the Sc-44 factor, 874MBq was measured, which corresponded to a 0.33% deviation from the reference activity.

#### 3.1.2. Vendor Method

In Table 4 and Table 5 below, the calculated factors for both scandium-43/44 and scandium-44 are reported, based on Equation (1).

### 3.2. BGcal, COV and Average Residual Lung Error

In Figure 3, *BGcal*, *COV* and average residual lung error obtained were plotted for the different radionuclides used and scatter correction (for the case of gallium-68, scandium-44 and scandium-43/44) for the NEMA phantom.

Figure 4 shows the cross-section of the reconstructed phantom obtained for the different radionuclides and reconstructions used. No noticeable difference was observed for the different radionuclides and reconstructions in terms of image noise and sphere recovery.

From the images obtained, it can be seen that the noise (in this study, represented by the *COV*) was similar for all radionuclides used (7.49%, 6.99%, 7.01%, 7.23%, 7.11%, 6.83% and 7.21%, respectively for fluorine-18, gallium-68 with *rSC*, gallium-68 *aSC*, scandium-43/44 *rSC*, scandium-43/44 *aSC*, scandium-44 *rSC* and scandium-44 *aSC*). The differences observed were the higher average residual lung error obtained for the reconstructions using *aSC*, the larger error in the background calibration in the scandium-43/44 with the absolute scatter and scandium-44 reconstructed with the *rSC*. Although showing greater differences, all residual lung error differences were <13%. The background calibration showed the largest differences for the scandium-43/44 mixture with *aSC* and radioisotopically pure scandium-44 with *rSC* at >20%, while the other radionuclides and scatter corrections indicated differences below the 10% recommended limit for [^18^F]FDG.

### 3.3. Recovery Coefficient

In Figure 5a,b, recovery coefficient mean and max were plotted for the NEMA phantom and the different radionuclides.

## 4. Discussion

In this study, the capability of a commercial PET device was evaluated to quantify a defined mixture of scandium-43/scandium-44 compared with quantitative PET achievable with pure scandium-44, fluoride-18 and galium-68.

The first important aspect when using non-standard radionuclides is capability of the local dose calibrators and PET/CT devices to provide reliable quantitative assessments. For the dose calibrators, Lima et al. [7] used a similar approach of defining local calibration factors based on a linearity method for scandium-44. The authors also compared their measurements to other quantitative approaches found in the literature [3,6,16], which assumed the scandium-44 measurement as a product of the measurement taken with the fluorine-18 setting multiplied by a factor (0.8). In this study, this linearity method was utilized, along with a vendor-technetium-99m reference-based method and, as per Lima et al. [7], it was compared with the fluorine-18-setting method. Using both linearity and vendor-recommended methods, a local dose calibrator factor was obtained for the scandium-43/44 mixture (factor 835 scale 1) and radioisotopically pure scandium-44 (factor 867 scale 1). Both local dose calibrator factors were found to be <1% from the reference activity, with the scandium-43/44 mixture at 0.89% and the scandium-44 at 0.33%. Based on Lima et al. [7], using the factor (factor 760 scale 0.56) on a similar dose calibrator obtained a difference of −1.19%. The difference between the factors employed (including factor and scale) was due to the software version available at the different institutions. The latest software version only allowed the scale to be either 1.0 or 0.5. Based on the factor used for fluorine-18 measurement and multiplying it by 0.8 [3,6], a difference of 40% was obtained. The linearity method and the vendor-technetium-99m reference-based method indicated lower differences and, as a result, were chosen for all subsequent activity dispensing.

With regard to the PET/CT capabilities, the device quantification (background calibration—*BGcal*), noise (*COV*) and effect of scatter corrections were initially evaluated, as well as in the average residual lung error. For the *COV*, with similar activity ratio between spheres and background (Table 3), all radionuclides presented similar noise levels: 7.49%, 6.99%, 7.01%, 7.23%, 7.11%, 6.83% and 7.21%, for fluorine-18, gallium-68, with *rSC*, gallium-68 with *aSC*, scandium-43/44 with rSC, scandium-43/ 44 with *aSC*, scandium-44 with *rSC* and scandium-44 with *aSC*, respectively. All levels were <15%, as recommended for fluorine-18 [11,12,13,14,17].

Regarding the *BGcal*, based on the acceptable deviation of 10% between measured and known background activity concentration [17], only scandium-43/44 with *aSC* and scandium-44 with *rSC* were outside tolerance levels. Interestingly, in contrast to Lima et al., the Biograph Vision produced acceptable results for scandium-44 when using *aSC* compared to their results with the Biograph mCT. As discussed by Lima et al. and confirmed by the current results (Figure 3), relative *SC* overcorrects the scatter contribution, due to the presence of the high-energy γ-radiation (Table 2), which results in a lower background activity concentration in the reconstructed image with respect to the known activity concentration. In contrast to the mCT, the differences in technology present in the Biograph Vision (which includes improved time resolution of 220 ps for the Vision compared to 550 ps for the mCT and narrower energy window with upper-level threshold ULD = 585 keV) compared to mCT (ULD = 650 keV)), contributed to this superior quantification and improved *SC*. Overall, using the optimal *SC*, a *BGcal* below 10% was achieved with 2.7%, 3.1%, 8.0% and 2.9% for fluorine-18 and gallium-68 with *rSC*, for scandium-43/44 with *rSC* and for scandium-44 with *aSC*, respectively.

Additionally, the impact of these corrections was evaluated in the lung region of the phantom. Knowing that there is no activity in the lung region, measurements of average residual lung error provide an assessment on the impact of the scatter and attenuation corrections in the final image. As shown previously in the *COV* results, *aSC* tended to under correct for scatter events. This was also observed in the average residual lung error, where all reconstructions with *aSC* showed an increased lung error in comparison with the *rSC*. Notably, even for the worst results, which were obtained for the scandium-43/44 mixture with *aSC*, the average residual lung error was <15%, which is comparable to acceptable performance for fluorine-18 [18].

As a final parameter, the recovery coefficients obtained for all radionuclides were investigated. Both *RC_mean_* and *RC_max_* were superior when using fluorine-18 compared to the other radionuclides. This can be explained by the pure positron emissions and corrections tailored to this radionuclide. Comparing the scandium-43/44 mixture to pure scandium-44, the RC results were comparable to gallium-68—a more widely used radionuclide.

Although having differences in phantom preparation and data analysis between the current measurements and NEMA or EARL, namely, the activity concentration between background and lesions, it is felt that the comparison is still meaningful for *COV* and *RC*s.

## 5. Conclusions

In this work, the PET/CT quantification capabilities for scandium-43/44 mixture in relation to radioisotopically pure scandium-44, fluorine-18 and gallium-68 were evaluated using a high-end SiPM PET/CT device. It was demonstrated that it could be achieved with either a mixture of scandium-43/44 or pure scandium-44 using a SiPM similar quantitative performance to clinically used radionuclides such as gallium-68 and fluorine-18. This situation will not only enable the clinical translation of scandium-44; it also raises hopes that scandium-43 can be used clinically, in spite of the fact that it cannot easily be prepared in radionuclidically pure form. Scandium radioisotopes benefit from longer half-lives than conventionally used PET nuclides such as gallium-68 and fluorine-18, which may benefit from clinical evaluation where a scan start can occur at later time points when background activity is already cleared. Moreover, scandium may be used for prospective dosimetry prior to therapy with ^177^Lu or ^90^Y-labeled counterparts. Importantly, delivery of scandium-44 and scandium-43 to PET centers without a cyclotron will be feasible over long distances and, thus, be of particular interest in countries in which the hospitals are far away from the production centers. The final optimization step refers to the use of a mixture of scandium-43/44 versus the use of scandium-44 only, as it can add to the reduction of an unnecessary dose to the producing personnel as well as the patient. Based on the current investigation, there is no obvious disadvantage of using a mixture of scandium radioisotopes. Hence, the concept could be translated to clinics without concern.

## Figures and Tables

**Figure 1 diagnostics-11-01826-f001:**
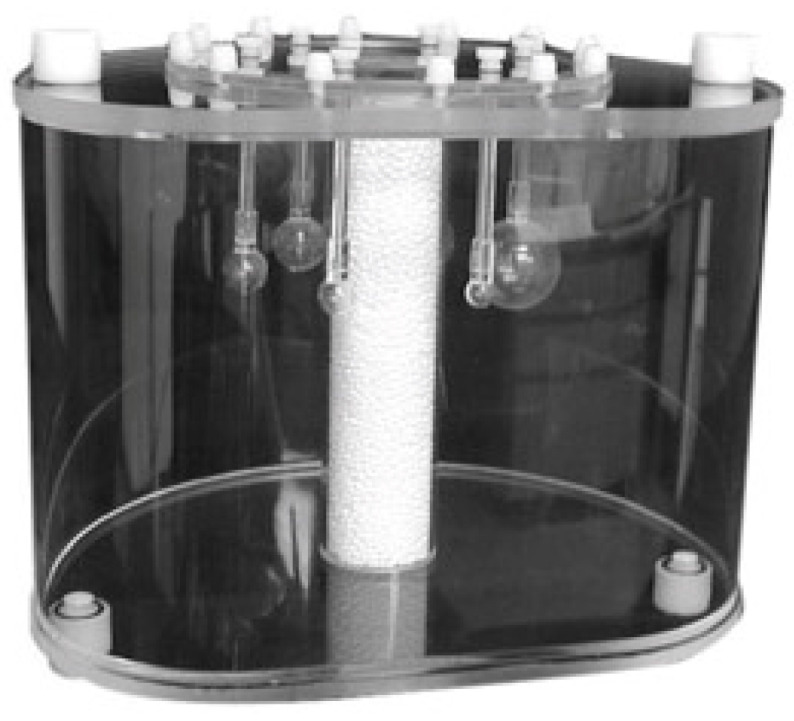
NEMA image-quality phantom with a lung insert, and spherical lesions of known dimensions.

**Figure 2 diagnostics-11-01826-f002:**
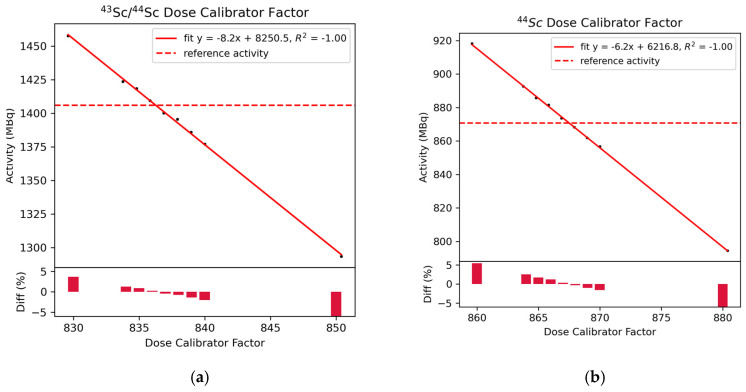
Results obtained using the linearity method for both the scandium-43/44 mixture and the radionuclidically pure scandium-44. (**a**) Measurements for the scandium-43/44 mixture across a range of calibrator factors; (**b**) Measurement for scandium-44.

**Figure 3 diagnostics-11-01826-f003:**
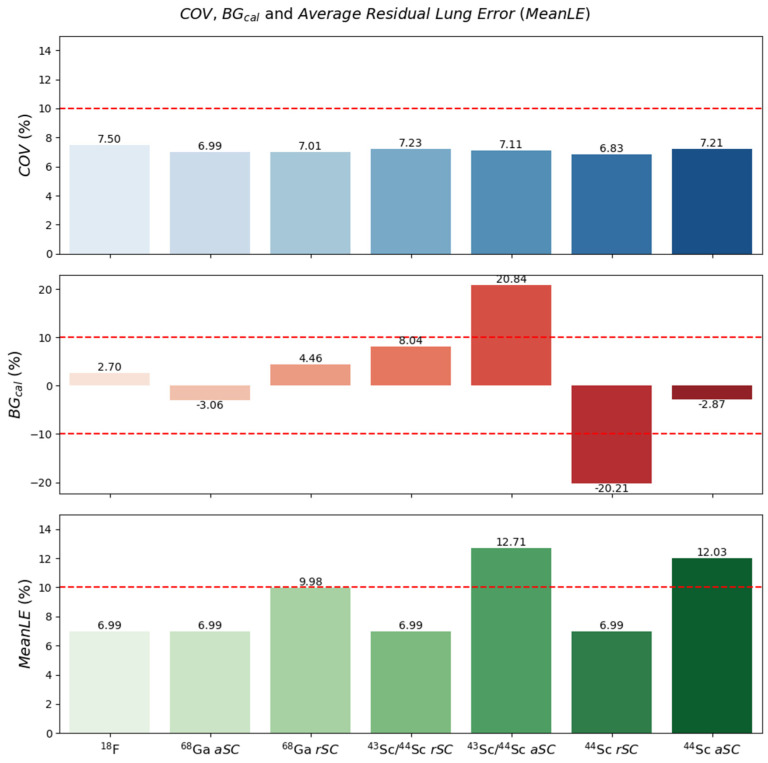
Results obtained for NEMA phantom. Background calibration (*BGcal*), *COV* and average residual lung error (*MeanLE*) in the NEMA phantom for fluorine-18, gallium-68, (*rSC* and *aSC*), the scandium-43/44 mixture (*rSC* and *aSC*) and pure scandium-44 (*rSC* and *aSC*), respectively.

**Figure 4 diagnostics-11-01826-f004:**
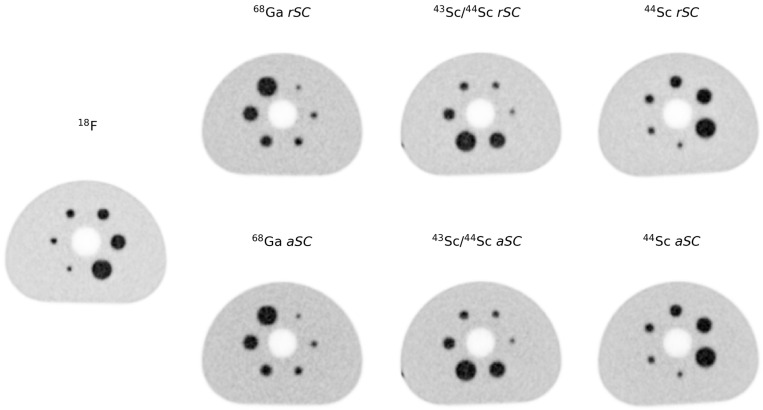
Reconstructions obtained for NEMA phantom. Cross-section images of the phantom using different radionuclides and reconstructions.

**Figure 5 diagnostics-11-01826-f005:**
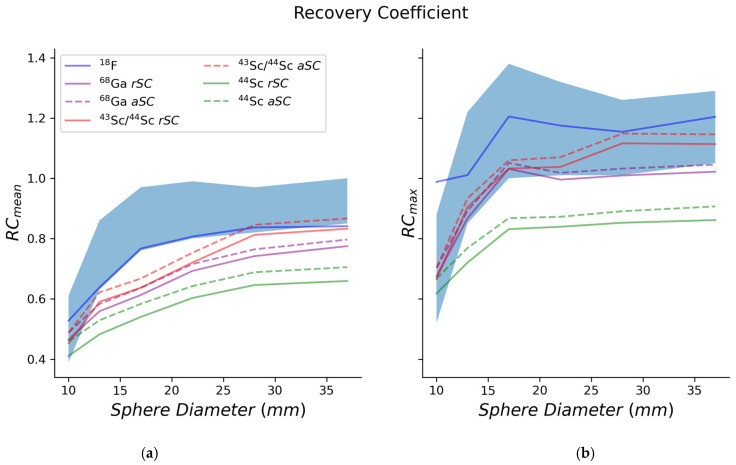
Mean (**a**) and maximum (**b**) recovery coefficients (*RC*s) for the NEMA phantom. The EARL *RC* ranges for [^18^F]FDG PET were used for comparative purposes.

**Table 1 diagnostics-11-01826-t001:** NEMA IQ Phantom characteristics.

	NEMA Phantom
Fillable volume (mL)	9400
Sphere diameters (mm)	10, 13, 17, 22, 28, 37
Lung insert (diameter/length mm)	50/180

**Table 2 diagnostics-11-01826-t002:** Physical decay characteristics of the radionuclides investigated.

	^43^Sc	^44^Sc	^18^F	^68^Ga
Half-life (h)	3.89	3.97	1.83	1.13
Decay method	EC, ß+/photon	EC, ß+/photon	EC, ß+/photon	EC, ß+/photon
ß + [% emissions]	88	95	97	89
E_ß + MAX_ [MeV]	1.2	1.47	0.634	1.9
E_gamma_ [KeV]	372.8 (23%)	1157.0 (99%)		1077 (3%)
h_10_ [(mSv/h)/GBq] at 1 m	0.174	0.324	0.160	0.149

EC—electron capture, h_10_—ambient dose equivalent rate.

**Table 3 diagnostics-11-01826-t003:** Concentration of the radionuclides investigated.

	^43^Sc/^44^Sc	^44^Sc	^18^F	^68^Ga
NEMA bkg concentration ^1^	4.70	5.88	4.43	4.99
NEMA sphere concentration ^1^	25.90	33.42	26.83	23.30
Ratio NEMA phantom	5.51	5.68	6.05	4.67

^1^ Concentrations are in kBq/mL.

**Table 4 diagnostics-11-01826-t004:** Results obtained for the scandium-43/44 mixture vial.

Factor	Scale	Radionuclide	Measured Activity ^1^ [MBq]	Deviation [%]
762	1	^18^F	1987	41.3
236	1	^99m^Tc	6229	343.
835 *calculated factor*	1	^43^Sc/^44^Sc	1419	0.89

^1^ Decay-corrected activity to reference time.

**Table 5 diagnostics-11-01826-t005:** Results obtained for the scandium-44 vial.

Factor	Scale	Radionuclide	Measured Activity ^1^ [MBq]	Deviation [%]
762	1	^18^F	1525	75.2
236	1	^99m^Tc	4748	445
760	0.56	^44^Sc [REF]	860	−1.19
835 *calculated factor*	1	^44^Sc	874	0.33

^1^ Decay corrected activity to reference time.

## Data Availability

Data available on reasonable request.

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
