# Peer review of "Fifty Shades of Scandium: Comparative Study of PET Capabilities Using Sc-43 and Sc-44 with Respect to Conventional Clinical Radionuclides"

_diagnostics, 2021, doi:10.3390/diagnostics11101826_

Round 1

Reviewer 1 Report

Lima et al presented a very interesting study to demonstrate that the purification of scandium production is not demanding from the imaging quality point of view. This is very inspiring and may promote the translation of the new radioisotope in clinic. In general, this study is well designed and well performed. The manuscript has high quality. It would be nice if the authors could clarify two more questions,

1. I am not sure if the mixture of Sc43 and Sc44 has constant proportion? As we know, the calibration is important for quantification. It would be nice if the calibration can be constant in practice.

2. I am not sure if the mixture of Sc43 and Sc44 has consequences in biological clearance? 

Author Response

  1. I am not sure if the mixture of Sc43 and Sc44 has constant proportion? As we know, the calibration is important for quantification. It would be nice if the calibration can be constant in practice.

We thank the reviewer for this important remark. The proportion of Sc-43 to Sc-44 will be constant at a given energy the target is irradiated at a cyclotron and this will depend on the cyclotron used. The supplier ought to be able to provide the ratio by measurement via gamma spectrometry after production. We are currently planning product assessment and calibration thereof with our metrology collaborators and hope to produce a paper in this regard in the near future.

With respect to the end-user, different proportions of Sc-43/Sc-44 is likely to impact on the calibration factor used, since dose calibrators use the measure gain multiplied by this calibration factor to establish the activity measured. Different proportions will have different ratios of emissions.

  1. I am not sure if the mixture of Sc43 and Sc44 has consequences in biological clearance? 

Sc-43 and Sc-44-labelled tumour targeting agents are chemically identical even if they differ slightly in their physical decay properties. The pharmacokinetics (uptake, distribution, metabolism, excretion) depends on the chemical structure of the tumor targeting agent and coordinated metal, but it is not relevant which radioisotope (it could also be Sc-47) is used. As a result, it does not matter whether a pure Sc-43 radioconjugate, a pure Sc-44 radioconjugate, or a mixture of both is used.

In the revised manuscript, this point has been clarified in the introduction section.

Reviewer 2 Report

Novel radionuclides are becoming of increasing interest as the field of theranostics grows. Scandium radioisotopes are of relevance in this context and improving the availability is a desirable outcome. This is a a very technical paper and not being a physicist myself, I don't feel that I can comment on the more technical aspects of the calculations performed but I think it would be helpful to clinicians to understand the consequences of the high energy gamma of Sc-43 on both image quality and radiation dosimetry and how these might be ameliorated by the mixture.

Author Response

Thank you for the feedback. We appreciate the interest of radionuclide development towards radiometals in theragnostics and our attempt to show their application in commercial imaging devices. The image quality of Sc-43 compared to Sc-44 and a mixture thereof is described in Domnanich et al. (DOI 10.1186/s41181-017-0033-9), while Sc-44 is compared to the more traditional imaging nuclides Bunka et al. (DOI 10.1016/j.apradiso.2016.01.006).

Reviewer 3 Report

The AA present a comparative study of PET capabilities using Sc-43 and Sc-44 with respect to conventional clinical radionuclides

The paper is interesting

Some points need clarifications/corrections:

  • Page 3, lines 79-84: a reference needs
  • Page 5, line 150: the formula is not complete
  • Page 6, line 184: where is Figure 3b?
  • Page 8, lines 195-199: this part is not clear
  • Page 8, line 208: ..coefficient max and mean… are inverted in the Figure 5
  • Figure 5 (b): RCmax values?
  • Discussion has to be reviewed
  • Page 10, line 250: Table 3 is Table 2

Author Response

  • Page 3, lines 79-84: a reference needs

We thank the reviewer for spotting the missing reference. It has been included in the revised manuscript.

  • Page 5, line 150: the formula is not complete

The reviewer is correct: there is an extra Eq6 in the manuscript. Initially, the RCmean and RCmax were separated in two formulae, but they were subsequently merged into Eq5. As a result, Eq6 was not needed.

  • Page 6, line 184: where is Figure 3b?

We thank the reviewer for pointing out this mistake. The liver phantom was removed from this study. So Figure 3 should be singular. This error has been corrected.

  • Page 8, lines 195-199: this part is not clear

This line was modified. The idea was to present the noise level for the different radionuclides in this study.

  • Page 8, line 208: ..coefficient max and mean… are inverted in the Figure 5

We thank the reviewer for highlighting this error. In the revised manuscript, the sentence has been rephrased.

  • Figure 5 (b): RCmax values?

Indeed, Figure 5 b is RCmax.

  • Discussion has to be reviewed

We thank the reviewer for this hint. The discussion was reviewed by a native English speaking co-author. The spelling mistakes were corrected, and table reference was corrected.

  • Page 10, line 250: Table 3 is Table 2

We thank the reviewer for highlighting this error. In the revised manuscript, the numbering of the tables was corrected.